

**A new approach for GNSS tomography from a few GNSS stations**
Nan Ding[1], Shubi Zhang[1], Suqin Wu[2], Xiaoming Wang[3], Allison Kealy[4], Kefei Zhang[2]
[1] School of Environment Science and Spatial Informatics, China University of Mining and
Technology, Xuzhou, China
[2] SPACE Research Centre, School of Mathematical and Geospatial Sciences, RMIT University,
Melbourne, Victoria, Australia
[3] Academy of Opto-Electronics, Chinese Academy of Sciences, Beijing 100094, China
[4] Geospatial Science, School of Science, RMIT University, Melbourne, Victoria, Australia
**Correspondence email**: metdingnan@163.com
**Abstract**
The determination of the distribution of water vapor in the atmosphere plays an important role in
the atmospheric monitoring. Global Navigation Satellite Systems (GNSS) tomography can be
used to construct 3D distribution of water vapor over the field covered by a GNSS network with
high temporal and spatial resolutions. In current tomographic approaches, a pre-set fixed
rectangular field that roughly covers the area of the distribution of the GNSS signals on the top
plane of the tomographic field is commonly used for all tomographic epochs. Due to too many
unknown parameters needing to be estimated, the accuracy of the tomographic solution degrades.
Another issue of these approaches is their unsuitability for GNSS networks with a few stations as
the shape of the field covered by the GNSS signals is in fact roughly an upside-down cone rather
than the rectangular cube as the pre-set. In this study, a new approach for determination of
tomographic fields fitting the real distribution of GNSS signals on different tomographic planes
at different tomographic epochs and also for discretization of the tomographic fields based on the
perimeter of the tomographic boundary on the plane and meshing techniques is proposed. The
new approach was tested using three stations from the Hong Kong GNSS network and validated
by comparing the tomographic results against radiosonde data from King's Park Meteorological
Station (HKKP) during the one month period of May, 2015. Results indicated that the new
approach is feasible for a three-station GNSS network tomography. This is significant due to the
fact that the conventional approaches cannot even solve a few stations network tomography.

**1 Introduction**

Information of the distribution and variation of atmospheric water vapor is essential for
meteorological applications. Nowadays, the most commonly used technology for measuring
atmospheric water vapor is radiosonde due to its high vertical resolution and high accuracy, even
though  its horizontal resolution is very low−several hundreds of kilometers, and its temporal
resolution is also low−twice daily. With the development of Global Navigation Satellite Systems
(GNSS),  using GNSS measurements to remotely sense water vapor in the atmosphere has



attracted significant attention due to their 24-hour availability, global coverage and low cost.
based on GNSS measurements collected from a regional or global GNSS reference network, a
regional or a global tomographic model, which is three-dimensional (3D), can be constructed.
The tomogpaphic model reflects the spatial variation of water vapor in the time period
investigated, thus it has the potential to be used to investigate the evolution of heavy rain events
for severe weather forcast (Wang et al., 2017; Chen et al., 2017; Zhang et al., 2015).
Using the slant wet delays (SWDs) estimated from the GNSS signals of a GNSS network
to construct a tomographic model is called GNSS tomography. Flores et al. (2000) built the first
GNSS tomographic model using $4 \times 4 \times 40$ voxels and developed Local Tropospheric Tomography
Software (LOTTOS) for simulation and processing of GNSS data. Gradinarsky (2002) developed
the wet refractivity Kalman filter (WeRKaF) for tomographic inversion of GNSS data and the
filter mainly focused on the initialization of the tomographic covariance matrix used in the
implementation of the Kalman filter. Troller et al. (2006) developed the atmospheric water vapor
tomography software (AWATOS) based on double-differenced GPS observations and double-
differenced phase residuals. Rohm and Bosy (2009) addressed the issue with the ill-condition of
tomographic equations using the Moore-Penrose pseudo inverse of the variance-covariance
matrix. In order to minimize the discretization effects, Perler et al. (2011) for the first time
proposed using node parameterization in GNSS tomographic modeling. Chen and Liu (2014)
optimized a water vapor tomographic region through moving voxel location along the latitudinal
and longitudinal directions until the number of the voxels that contain GNSS signals reached the
maximum. Yao et al. (2016) improved the utilization rate of GNSS observations in the modeling
by adding extra voxels on the top of the tomographic region where some satellite signals partly
cross the tomographic field. Ding et al. (2017) developed an access order scheme called prime
number decomposition (PND) for minimizing the correlation between the SWDs which are the
sample data of tomographic modeling. The above GNSS tomographic approaches were tested
using various numbers of GNSS stations, majority of which were a few tens of stations, and the
maximum and minimum were 270 and 8 respectively.
In all the above tomographic approaches, the tomographic fields are all assumed
rectangular cubes. The size and location of the rectangular cubes are determined based on the
distribution of GNSS signals only on the top boundary of the tomographic field−the rectangular
cube that best fits the top boundary is adopted (Bastin et al., 2005; Bender et al., 2009;
Champollion et al., 2005; Ding et al., 2017; Gradinarsky and Jarlemark, 2004; Hoyle, 2005;
Rohm et al., 2014; Seko et al., 2000; Troller et al., 2006; Xia et al., 2013; Ye et al., 2016). In fact,
the field that GNSS signals cover has a shape of upside-down cone, roughly, meaning that in the
part near the edge of the cube, especially in the lower part, none of the GNSS signals cross
through. This region is named empty spatial region (ESR) in this paper merely for convenience.
In fact, the inclusion of those voxels/nodes in the ESR in the discretization of the model not only
does little contribution to the improvement on the accuracy of the model solution but also adds
extra meaningless unknown parameters to be estimated. More parameters mean more horizontal
constraints are needed and also degradation of the accuracy and stability of the solution,
especially in the case the network consists of a few stations, e.g.  only three stations. This is
because the difference in the sizes covered by the GNSS signals in the bottom and top planes of
the tomographic field is large, meaning a large number of voxels/nodes in the ESR and far away
from the observed signals, especially in the lower part of the tomographic field. In the estimation
process of the model, the horizontal constraints imposed on these nodes/voxels are usually from
extrapolated results based on their nearest observations. If these voxels/nodes are far away from





the observed signals, the constraints are too weak and will cause difficulty in the solving of the
tomographic equations. The large number of nodes/voxels contained in ESRs stemming from a
small number of GNSS stations is the main reason for the unsuitable of the current GNSS
tomographic approaches to   using a-few-station networks.

In this study, a new node parameterization approach for dynamic determination of
tomographic fields and the discretization of the fields at each tomographic epoch was proposed.
It is adaptive node parameterization for varying density on different tomographic planes. This
differs from all current approaches in which the same pre-set rectangular cube roughly
determined by the distribution of the signals only on the top tomographic plane is adopted for all
planes and all epochs of the tomography. In addition, for the discretization of the tomographic
field determined for each plane at each epoch, the location and number of all the nodes on the
plane are determined according to the size of the tomographic field.  As a result, the tomographic
model is tailor-made for all planes and alll epochs. Moreover, the new approach is applicable to
GNSS networks with any number of stations, i.e. equal to or larger than three.
**2 Methodology**
2.1 Observations of GNSS Tomography

GNSS signals are bent and delayed when they propagate through the atmosphere. The
atmosphere can be divided into the ionosphere and troposphere. The ionospheric delay can be
cancelled out using an ionosphere-free linear combination of dual-frequency observations. The
tropospheric delay can be divided into two components−the dry delay and the wet delay. The wet
component is the SWD and can be expressed by
$$SWD = m_w(e) \cdot \left\{ ZWD + \left[ G_N^w \cdot \cos(\phi) + G_E^w \cdot \sin(\phi) \right] \cdot \cot(e) \right\} + R \qquad (1)$$
where $m_w(e)$ is a wet mapping function and the VMF1 mapping function was used in this study;
$G_N^w$ and $G_E^w$ are the wet delay gradients in the north–south and east–west directions, respectively;
$R$ is the unmodeled delay; $ZWD$ is the zenith wet delay of the GNSS station, which can be
obtained by subtracting the zenith hydrostatic delay (ZHD) from the zenith total delay (ZTD).
The ZHD can be calculated by a standard tropospheric model such as the most commonly used
Saastamoinen model (Saastamoinen, 1972) and the ZTD is estimated (as an unknown parameter)
in GNSS data processing;

In GNSS tomographic modeling, the SWDs of GNSS signals in a tomographic field are
used as the observations for the estimation of water vapor parameters in the field.
2.2 Tomographic modeling
2.2.1 General approaches

Voxel and node parameterization are the two common GNSS tomographic approaches. In
the former, the tomographic field, which is usually assumed as a rectangular cube, is divided into
many voxels (small rectangular cubes) and in the latter, and the field is discretized by nodes, as
all the black and circle nodes shown in Fig. 1. In this study, the node parameterization approach
was adopted due to its better fitting of the spatial correlation of water vapor.





In the current node parameterization approaches, if the GNSS network is very small, e.g.
a three-station network from the Hong Kong Satellite Positioning Reference Station Network
(SatRef) as shown in Fig. 1, a large number of nodes are in the ESR (see the hollow circles)
within the rectangular cube which is the tomographic field. These nodes, as part of the unknown
parameters, need to be estimated. The inclusion of these unknown parameters in the estimation
process does not only add more 'redundant' parameters but also degrades the accuracy of the
solution.


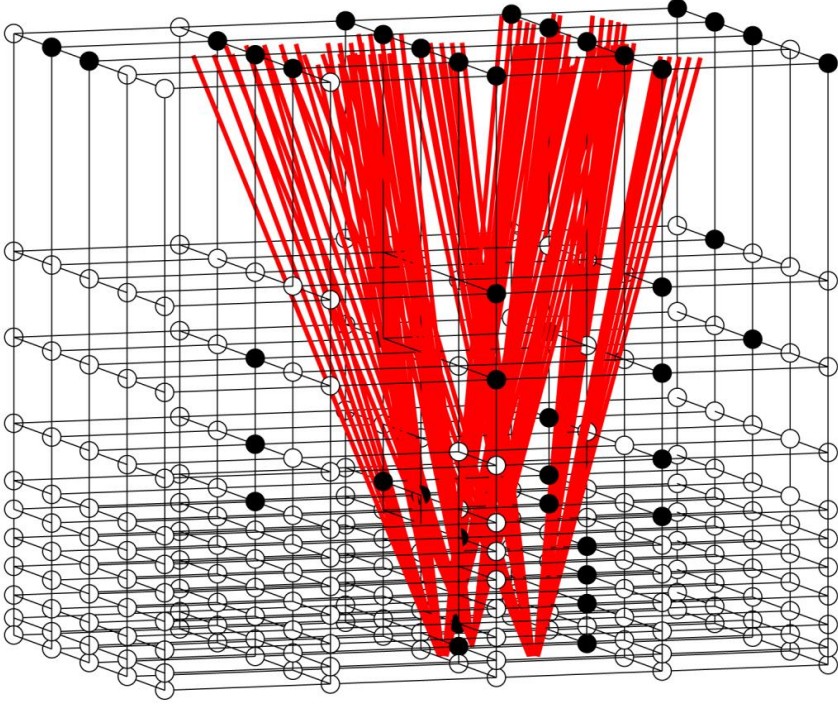


**Figure 1.** A three-station GNSS network from the Hong Kong Satellite Positioning Reference
Station Network (SatRef) as an example for GNSS tomography−the rectangular cube is the
tomographic field adopted in current node parameterization approaches, the solid nodes are those
near GNSS signals and the hollow nodes are those in the ESR.

In addition, a fixed rectangular cube is used as the tomographic field for all time in the
current approaches, In fact, the spatial region that the signals travel through varies with time, as
shown in Fig. 2 for the different distributions of the signals at the three stations shown in Fig. 1
on the top plane of the tomographic field at UTC 0 on 1 (day of year (DOY) 121), 16 (DOY 136)
and 31 (DOY 151) in May 2015.


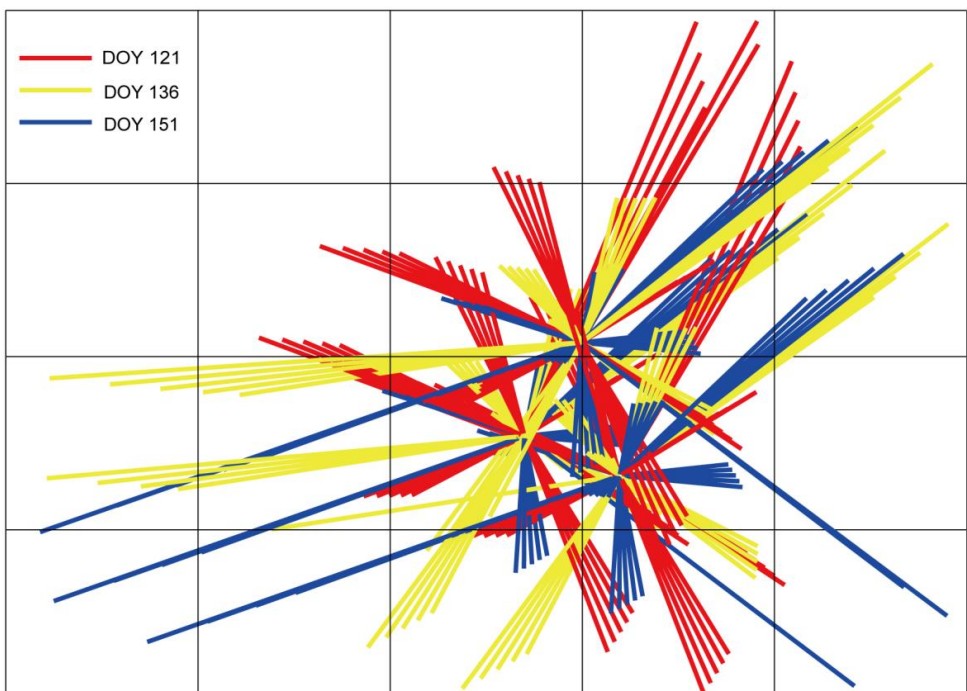


**Figure 2.** Distributions of GNSS signals at the three stations shown in Fig. 1 on the top plane of
the tomographic field at UTC 0 on 1(DOY 121), 16 (DOY 136) and 31 (DOY 151) in May 2015.

To address the above issues, a new node parameterization approach that dynamically
adjusts the tomographic field based on the spatial distribution of the GNSS signals at the
tomographic epoch and also dynamically adjusts the location and number of all the nodes based
on the size of the tomographic field is proposed. Its procedure is elaborated in the next section.

2.2.2 New approach

       The procedure for the new approach mainly includes two steps–determination of
tomographic field and determination of node position, which are introduced below.

*i) Determination of tomographic field*

       A tomographic field is regarded to be comprised of many layers in the vertical dimension
and these layers with the same or different thickness, depending on the distribution of water
vapor at the height of the layer, as shown in Fig. 3(a), each layer is formed by two neighboring
horizontal planes. After all these planes are determined, the next task is to determine the
tomographic boundary for each plane, according to the distribution of the GNSS signals on the
plane. Fig. 3(b) shows the tomographic boundary on each of the planes shown in Fig. 3(a), which
is determined from the following three steps that were used in the Graham scan (Graham, 1972)
determining all the intersections ( the blue points) of the GNSS signal paths on the plane (they





are name pierce points in this paper); 2) using a stack of the pierce points to detect and remove
all those pierce points that are in concavities; and 3) connecting the rest pierce points to form a
convex hull, which is the tomographic boundary (black polygon).

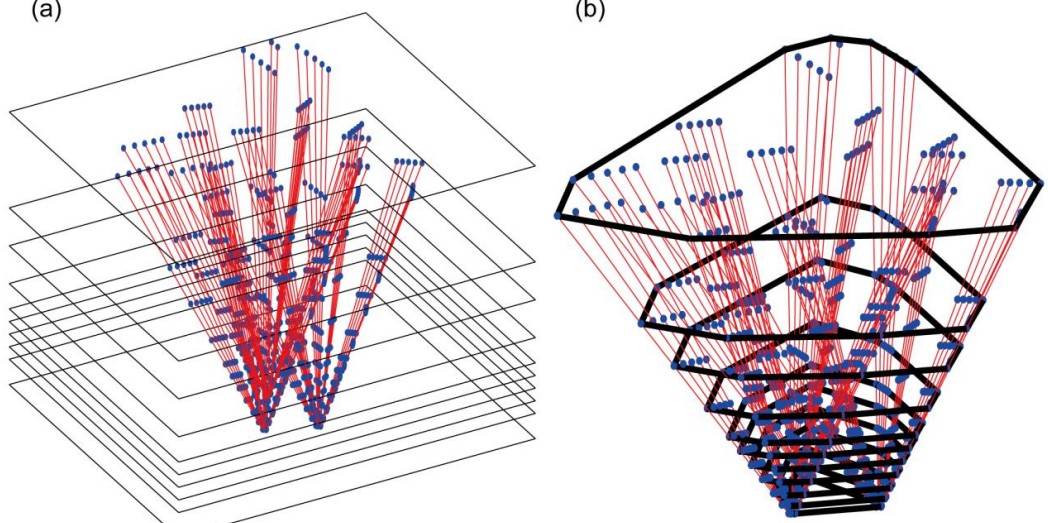

(a)                                              (b)


**Figure 3.** (a) A tomographic field is divided by many layers, the thickness of which is dependent
upon the distribution of water vapor in the layer, the red lines are the sampling GNSS signals and
the blue points are the intersections of the GNSS signals on each horizontal plane; and (b)
Tomographic boundary is depicted by the black polygon on each horizontal plane.

Since the shape of the tomographic boundary determined using the new approach is irregular,
it is difficult to generate equidistant nodes within the boundary. This differs from current node
parameterization approaches in which uniformly distributed nodes can be easily pre-set. In this
study meshing techniques are used to adjust the position of nodes for each plane and each
tomographic epoch, and their procedure is discussed in the next section.
*ii) Determination of node position*
Meshing techniques for the generation of equidistant nodes of a GNSS tomographic
model include three steps and each of the steps is introduced below.
1) A mesh background in a desired size with nodes is used to provide initial nodes for
each plane see Fig. 4(a) where the polygon is obtained from the last section for the tomographic
boundary on the plane and at all the vertices of the polygon a new set of nodes are also attached
to the initial nodes, see Fig. 4(b) for the final initial nodes.
2) Delaunay triangulation (Delaunay, 1934) is used to establish a topology for the above
initial nodes on each plane. It determines non-overlapping triangles that fill the region in a
polygon such that every edge is shared by at most two triangles and none of the vertices is inside
the circumcircle of any of the triangles. Delaunay triangulations maximize the minimum angle of



all the triangles to avoid sliver triangles which has undesirable properties during some
interpolation or rasterization processes (Edelsbrunner et al., 2000). Several methods have been
developed to compute the Delaunay triangulation such as the commonly used flipping edges and
conversing a Voronoi diagram. In this study, the flipping edges method is adopted to connect the
initial nodes shown in Fig. 4(b) by the edges of Delaunay triangles on each plane and the
topology formed is shown in Fig. 4(c).

3) The force displacement algorithm (Persson, 2005) is applied to the above topology for
the adjustment of the initial nodes into equidistance with a reasonable length fitting the size of
the tomographic boundary on each plane. This method is based on the assumption that each edge
in the topology has a force value (let it be $F_{ij}$) equal to the length of the edge. It can be used to
make all the edges' $F_{ij}$ close to the same and reasonable pre-set force value $F_0$ for a (roughly)
regularly distributed mesh. This is the main reason for the introducing of this method to this
study for adjusting the nodes in the irregular tomographic boundary (like Fig. 4(c)) into
equidistance (roughly). The force displacement algorithm is an iterative process as:

$$[X^k \quad Y^k] = [X^{k-1} \quad Y^{k-1}] + Scal \cdot [F_x^{\ k} \quad F_y^{\ k}] \tag{2}$$

where $X^k$ and $Y^k$ are the vectors of the x and y coordinates respectively of all the nodes on the
plane at the kth iteration and k-1 denotes the previous iteration; Scal is a relaxation factor for
constraining the amount of the movement from the k-1th iteration to an appropriate value, for
which a 0.2 value is commonly used; $F_x^{\ k}$ is the vector of the vector sums of all the forces
working on each of the nodes in the x direction, $F_y^{\ k}$ is that in the y direction.

After the above algorithm is performed, all the nodes on the plane can be adjusted from
the initial position (Fig. 4(c)) to equidistant position (Fig. 4(d)) through a series of iterations.

It is noted that the sizes of the tomographic boundaries on different planes are different
(Fig. 3(b)) while the numbers of the signals on different planes are the same, so the densities of
the signals on different planes are different, and the densities of the nodes on different planes
better be different through using different $F_0$ values. In this study, the $F_0$ value for the ith plane is
calculated by:

$$F_0^i = C \cdot mean(Ls^i) \tag{3}$$

where C is a constant coefficient and 0.68 is adopted for all planes; and mean($Ls^i$) is the mean of
all the lengths of the edges on the polygon.



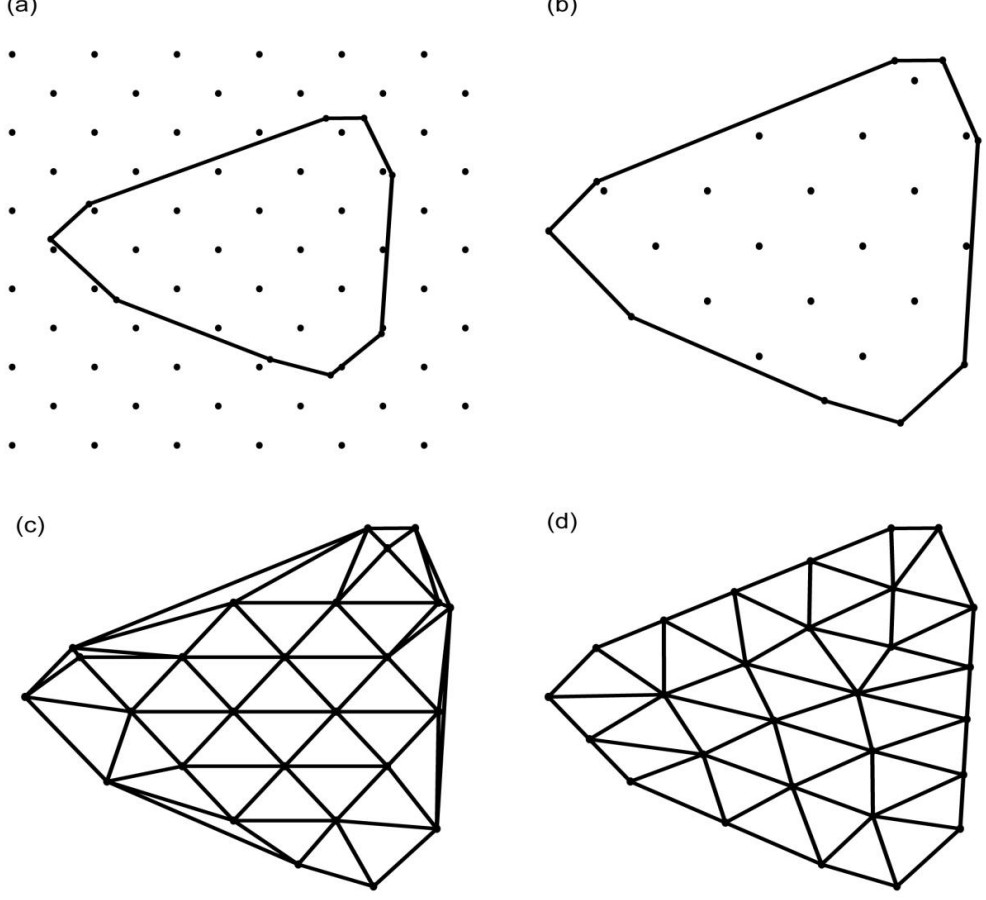

**Figure 4.** (a) Two sets of nodes for initialization−one set is generated using a mesh background
with a desired size which is usually slightly larger than the region of the GNSS signals at all time
and the other set is at all the vertices of the polygon (all black points); (b) Initial nodes; (c)
Topology formed using Delaunay triangulation; (d) Nodes with equidistance adjusted based on
the force displacement algorithm.

2.3 Observation equations

After equidistant nodes for all planes are determined (like Fig. 4(d)), the next step is to
estimate water vapor parameters at these nodes from observation equations of GNSS-derived
*SWDs*. The derivation of the observation equations is as follows.

Theoretically, *SWD* is defined as the integral of wet refractivity $N_w$ along the signal path $s$

$$SWD = 10^{-6} \cdot \int_s N_w ds \qquad (4)$$

It can be further decomposed into integrations of $n$ layers:





$$SWD = \sum_{i=1}^{n} \int_{s_i}^{s_{i+1}} N_w^s(i) ds = \sum_{i=1}^{n} SWD_i \qquad (5)$$

where $N_w^s(i)$ is wet refractivity in the $i$th layer ; $s_i$, $s_{i+1}$ are the start and end points of the
layer/integral; and $SWD_i$ is the part of the $SWD$ in the $i$th layer
In GNSS tomography, in each of the piecewise integrals expressed in Eq. (5), i.e. $SWD_i$,
the signal path in the layer is further divided into several equally spaced points and then $SWD_i$ is
approximated as a function of wet refractivity at these points using the Newton-Cotes formulae
(Perler et al., 2011). In this study, $SWD_i$ is approximated by the Newton-Cotes formulae of 4
degree at five equally spaced points, as (P $_1$…, P $_5$) shown in Fig. 5 where the plane $i$ and plane
($i$+1) are the two horizontal planes corresponding to the above $s_i$, $s_{i+1}$ respectively, and the black
solid dots denote some of the equidistant nodes obtained from Fig. 4(d).
The methods for obtaining wet refractivity at each of the points are as follows.
i) Wet refractivity at points $P_1$ and $P_5$ (which are on the $i$th and ($i + 1$)th planes
respectively) can be calculated using the interpolation method of the inverse-distance-weighted
(IDW) mean of the sample wet refractivity data from its surrounding nodes:
$$P_{wet} = \frac{\sum_{j=1}^{m} w_j \cdot n_j^{wet}}{\sum_{j=1}^{m} w_j} \qquad (6)$$

where $j$ is the index of the sample data, and $w_j$ is its weight determined by the inverse-distance;
and $m$ is the number of the sample data.
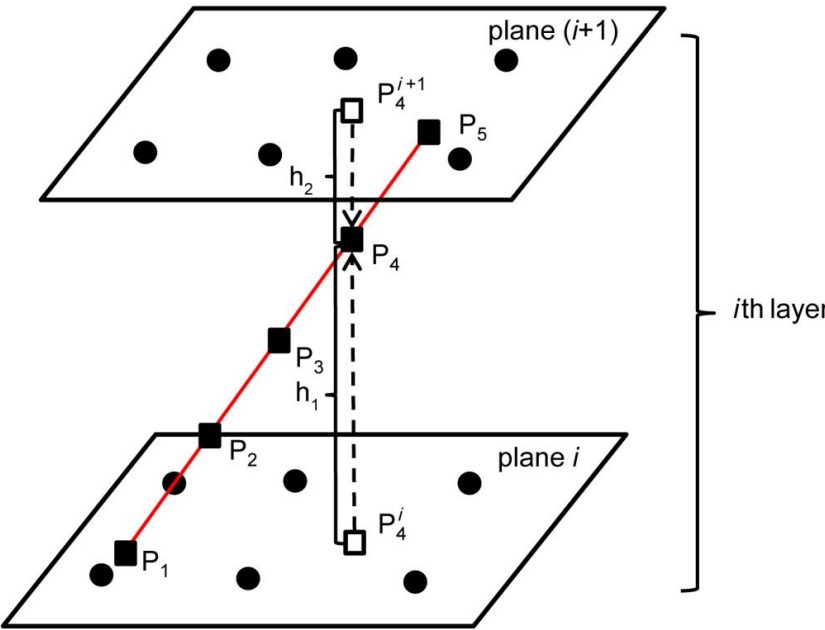





**Figure 5.** Five equally spaced points (black solid squares) for an approximation of wet
refractivity for the $i$th layer. $P_4^i$ and $P_4^{i+1}$ (black hollow squares) are the projected points of $P_4$ on
the $i$th and ($i+1$)th planes respectively, $h_1$ is the height difference between $P_4$ and $P_4^i$ , and $h_2$ is
that between $P_4$ and $P_4^{i+1}$ .

ii) Wet refractivity at points $P_2$ , $P_3$ and $P_4$, cannot be directly interpolated like that for $P_1$
and $P_5$, the following three-step procedure needs to be performed ($P_4$ is taken as an example): 1)
the position of $P_4$ is projected onto both the $i$th and ($i+1$)th planes to obtain two projected points
named $P_4^i$ and $P_4^{i+1}$, respectively; 2) the above interpolation procedure for $P_1$ and $P_5$ is used to
obtain wet refractivity $P_{4wet}^i$ and $P_{4wet}^{i+1}$ at $P_4^i$ and $P_4^{i+1}$ respectively; and 3) $P_{4wet}^i$ and $P_{4wet}^{i+1}$ are used to
obtain a weighed mean wet refractivity for the position of $P_4$ using [Reitan, 1963; Tomasi, 1981]:
$$P_4 = \frac{|h_1|}{(|h_1|+|h_2|)} P_{4wet}^i \cdot e^{-h_1/H} + \frac{|h_2|}{(|h_1|+|h_2|)} P_{4wet}^{i+1} \cdot e^{-h_2/H} \qquad (7)$$

where $h_1$ is the height difference between $P_4$ and $P_4^i$ and $h_2$ is that between $P_4$ and $P_4^{i+1}$ ; and $H$ is
water vapor scale height, which can be calculated by Tomasi [1977]:
$$H = \frac{10W}{\rho_s} \qquad (8)$$

where $W$ and $\rho_s$ are the vertical total water vapor content (in g m$^{-2}$) and surface humidity (in g m$^{-3}$)
respectively, and both can be obtained from GNSS data.

After the above procedures are carried out, $SWD_i$ can be expressed as a function of wet
refractivity at a set of nodes. This procedure needs to be performed for all $SWD_i$ ($i=1,2,..n$), then
the next step is to substitute these $SWD_i$ expressions and the $SWD$ observation into Eq. (5), to
form its GNSS tomographic observation equation.

The final GNSS tomographic observation equations of all SWDs from the GNSS network
for the tomographic modeling is expressed as:
$$A \cdot X = b \qquad (9)$$

where $A$ is the coefficient matrix of the model; $b$ is the vector of the SWD observations; and $X$ is
the vector of the wet refractivity parameters at all nodes.

The $X$ vector in Eq. (9) can be estimated using the least squares method. However, due to
the problem with the sparseness of $A$, the algebraic reconstruction technique (ART) was used to
estimate $X$ in this study.
2.4 Tomographic solution

The ART has been successfully applied to reconstruction of water vapor field (Chen and
Liu, 2014; Bender et al., 2011). Its main advantage is the high numerical stability, even under
adverse conditions and also relatively easy to incorporate prior knowledge into the reconstruction
process. The ART used to solve Eq. (9) is (Kaczmarz, 1937):
$$x^{k+1} = x^k + \lambda \frac{b_i - \langle a_i, x^k \rangle}{\|a_i\|_2^2} a_i \qquad i = 1, 2, \cdots, m \qquad (10)$$



where $a_i$ and $b_i$ denote the $i$th rows in $A$ and $b$ respectively; $x_k$ is the $k$th iterative solution; and $\lambda$ is a relaxation factor and the value of 0.2 was selected in this study.

It is noted that Eq. (9) needs to be sorted in a certain sequence for Eq. (10). This is different from the commonly used observation equation system in which the order of the observation equations is not a matter. In this study, an access order scheme based on prime number decomposition (PND) proposed in (Ding et al., 2017) was used for the ordering of the observation equations such that the observation equations between two consecutive iterations are largely uncorrelated.

The unknown parameters $X$ solved from Eq. (10) are the wet refractivity values at all tomographic nodes. In some meteorological applications, water vapor density may be preferred, in this case $X$ needs to be converted using a conversion factor $\Pi$ which is a function of water-vapor-weighted-mean temperature $Tm$ (Bevis et al., 1994; Wang et al., 2016) at the position of the nodes.

## 3 Test results

3.1 Data selection and tomographic scheme

Test data used in this study were from three stations in the Hong Kong Satellite Positioning Reference Station Network (SatRef), and the horizontal and vertical distributions of the three stations are presented in Fig. 6(a) and Fig. 6(b), respectively. The area of our interest ranges from 113.749 °E to 114.474 °E in the longitudinal direction, from 22.115 °N to 22.651 °N in the latitudinal direction and from 0 to 10800 m in the vertical direction. Radiosonde data from King's Park Meteorological Station (HKKP) (the blue triangle shown in Fig. 6(a) were used as the reference for the validation of our test results.

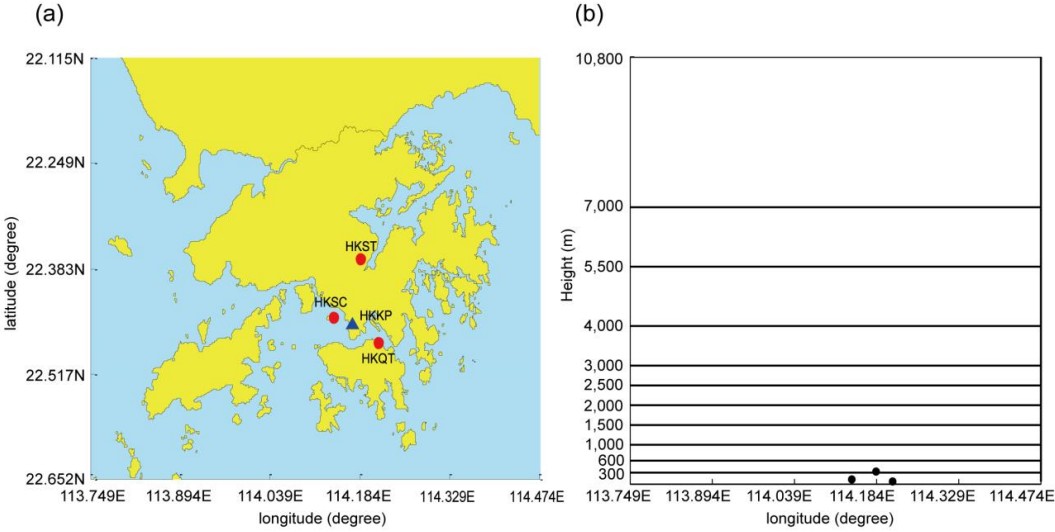

**Figure 6.** (a) Horizontal distribution of the three stations selected from the Hong Kong reference stations (red dots) and HKKP (blue triangle); and (b) Vertical distribution of the three stations (black spots) and vertical layers used in tomographic modeling.

The test data were from the whole month of May, 2015 (day of year (DOY) 121−151)
with the sampling rate of 30 seconds, and the GAMIT software was used to obtain SWDs at the
same rate in the data processing. For the tomographic modeling, a 5-minute sampling rate for
SWDs and a 30-minute interval for a tomographic epoch were adopted, meaning that the number
of SWD observations for a tomographic epoch was seven−including the two sample data at the
two ends of the interval. The reason for the selection of data from May 2015 is that its monthly
total rainfall was 513.0 mm, a 68% larger than the normal level of 304.7 mm.
The tomographic scheme for testing is as follows. The first step is to determine the
vertical planes/layers for the tomographic field.  Non-uniform vertical intervals from 300 to 3800
m (Fig. 6(b)) were selected for adaption to the inherent characteristic of water vapor spatial
distribution−it exponentially decreases with the increase of height.  The use of this structure can
also avoid too many unknown parameters overfitting the SWD observations. The next step is to
determine the tomographic polygon/boundary on each of the above planes using the methods in
section 2.2.1 and based on the GNSS signals in the tomographic interval, then according to the
polygon's perimeter, a $F_0$ value in the force displacement algorithm for determination of the
density of nodes on each plane is calculated. All $F_0$ results in our test are in the range of about
1800−10000 m corresponding to the range of height 300–10800 m. The position of the nodes on
each plane is determined by Eq. (2).
Figure 7 shows the boundary and nodes on three tomographic planes at tomographic
epoch UTC 0 on DOY 121, 2015 for an example. Tomographic model results are presented in
the next section.
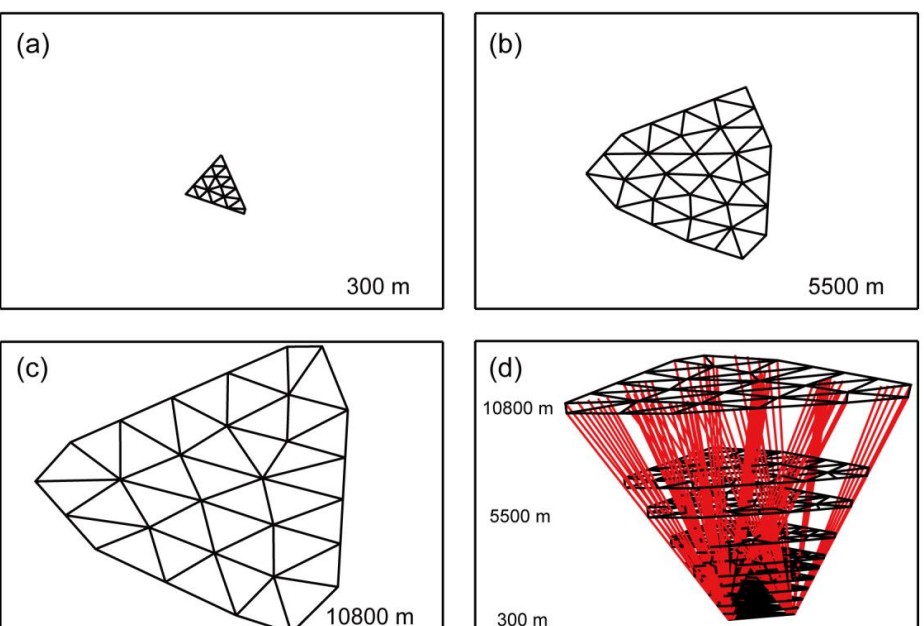

**Figure 7.** Tomographic boundary and nodes on three planes ((a), (b) and (c)) and the
tomographic field and nodes (d) at tomographic epoch UTC 0 on DOY 121, 2015.



### 3.2 Results of profiles


Water vapor density values obtained from the tomographic models at tomographic
epochs UTC 0 and UTC 12 on each day of the month (DOYs 121−151) were compared against
radiosonde (RS) data for evaluation of the model's accuracy.  The values of the tomographic
results at all RS sampling points were calculated first using the interpolation method mentioned
in section 2.2.1, then the root mean square error (RMSE) of the differences between the
interpolated values and RS observations at all the sampling points of the RS profile from the
ground surface to 10800 m at each epoch was calculated for the accuracy of the profile. All the
results at the 62 epochs during the 31-day period are shown in Fig. 8.

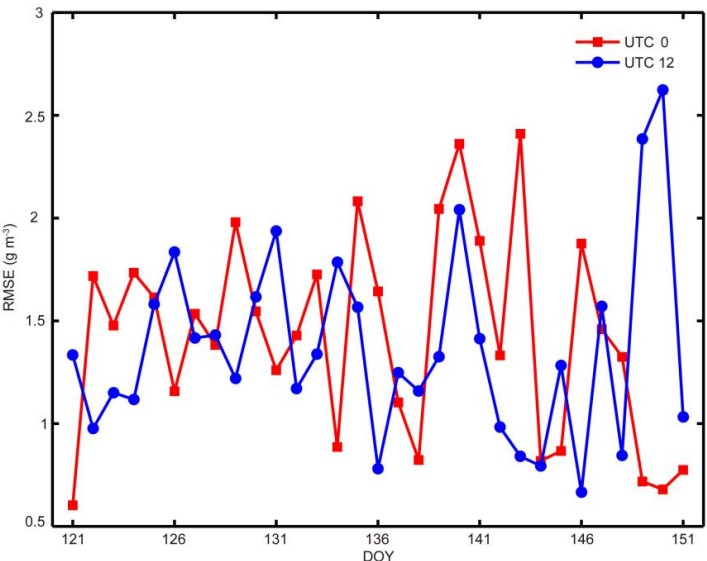



**Figure 8.** RMSE of model-derived water vapor density values at all RS sampling points of the
RS profile below 10800 m at tomographic epochs UTC 0 and UTC 12 on each day of the month
(DOYs 121−151).


The maximum RMSEs, i.e., the worst results,  at UTC 0 (red) and UTC 12 (blue) are on
DOY 143 and DOY 150 respectively; while the best result (the minimum RMSEs) at the two
epochs are on DOYs 121 and 146. In order to find the reason for the large difference between the
worst and best results, the tomographic field, the distribution of the signals and the nodes at these
four epochs are given in Fig. 9, where Fig. 9(a) and Fig. 9(b) correspond to the best results at
UTC 0 and UTC 12 respectively, both of which show uniform distributions of the GNSS signals.
However, the distributions of the GNSS signals corresponding to the worst results at UTC 0 (Fig
9(c)) and UTC 12 (Fig 9(d)) are different in the sparse signals shown in blue lines, which is the
reason for the poor accuracy of the model results.






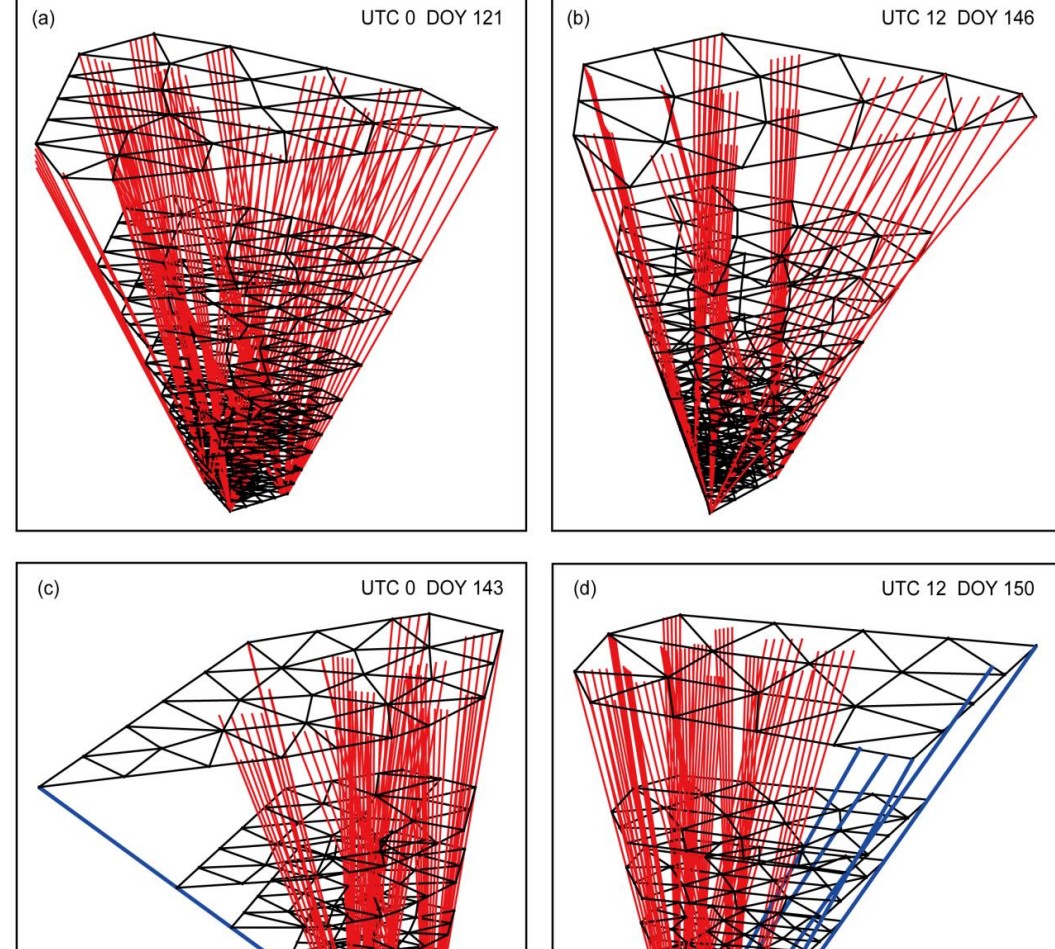

**Figure 9.** Tomographic field and signal distribution at tomographic epoch UTC 0 on DOY 121
(a), UTC 12 on DOY 146 (b), UTC 0 on DOY 143 (c), and UTC 12 on DOY 150 (d).

The results shown in Fig. 8 are the statistics of the model results for each epoch on each
day. The statistics of the model results at both epochs together in the whole month are presented
in Table 1. The three values listed in the table are all small, meaning that the new approach for a
few GNSS stations, such as three stations, is feasible.



**Table 1.** Monthly statistics of tomographic modeling results

| Statistic | RMSE (g m$^{-3}$) | Bias (g m$^{-3}$) | IQR (g m$^{-3}$) |
|---|---|---|---|
| | 1.477 | 0.239 | 1.430 |


To indicate the spread of all the errors (the ones used to calculate the above monthly
statistics), scatter plots shown in Fig. 10(a) are used to analyze the characteristics of these errors
in different intervals. The x and y axes denote the RS observation and the model result (in g m$^{-3}$)
respectively; each hollow circle corresponds to a sampling point's result; and the red line
represents  the "perfect" results, i.e. the model results equal to the RS results. Those hollow
circles that are on the red line have an error value of zero, those above the red line have a
positive error value, and the rest have a negative error value. The closer a hollow circle to the red
line, the smaller its error value.
How well all the hollow circles "fit" the red line indicates the overall quality of the model
results. It is clear that the hollow circles have a cigar-shaped (fusiform) distribution. The hollow
circles in both ending intervals ([0−5] and [20−25] g m$^{-3}$) more concentrate around the red line
than those in the middle part ([5−20] g m$^{-3}$). The reason for this is 1) most of the sampling points
in the [20−25] g m$^{-3}$ interval are located near the ground surface, where water vapor density
decreases exponentially with the increase of height and the density of the GNSS signals is very
high, resulting in relative high accuracy; 2) most of the sampling points in the [5−20] g m$^{-3}$
interval are located in the mid-height of the tomographic field, where the GNSS signals are
sparser than the [20−25] g m$^{-3}$ interval, leading to  a larger tomographic field, which results in a
lower accuracy; and 3) most of the sampling points in the [0−5] interval are located in the top
section of the tomographic field, where the water vapor values are smaller than the other two
intervals, leading to the smallest errors.

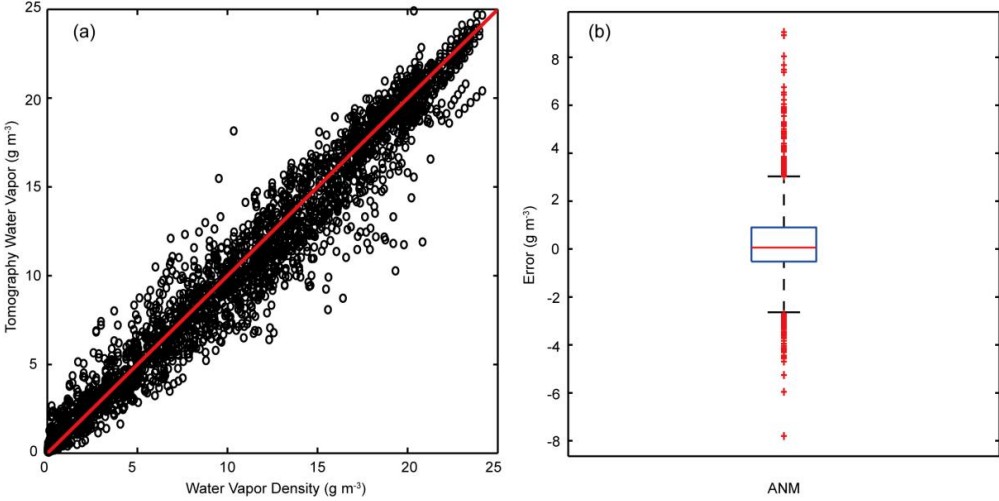






**Figure 10.** Graphic presentation for the distribution of the tomographic results at the two epochs on every day during the month: (a) scatter plot of water vapor density; and (b) box plot for outlier detection of the tomographic errors.

The box plot is mainly for the indication of those large errors at all sampling points. Q1 and Q3, which are the first and third quartiles respectively, determine the IQR value in Table 1; Q2, the second quartile, roughly reflects the bias of all the errors; the whiskers, i.e. the two black bars, located at Q1−1.5(IQR) and Q3+1.5(IQR), are for the determination of the lower and upper bounds of the criteria for outlier detection, e.g. the red cross marks are regarded outliers. Table 2 lists all the above characteristic values.

**Table 2.** Characteristic values of the box plots in Fig. 10(b).

| Statistic | Q1 (g m$^{-3}$) | Q2 (g m$^{-3}$) | Q3 (g m$^{-3}$) | Upper bound (g m$^{-3}$) | Lower bound (g m$^{-3}$) | Number of outliers |
|---|---|---|---|---|---|---|
| | −0.527 | 0.062 | 0.903 | 3.048 | −2.672 | 159 |

3.3 Results of different layers

In the last section, the RMSE of model-derived water vapor density values at all sampling points for each profile (Fig. 8) and the errors at all the sampling points and two epochs on each day during the month (Fig. 9) are analyzed for the assessment of the overall performance of the models. In this section, the monthly RMSE at all the sampling points but in 11 different tomographic layers and the monthly mean of the relative errors in these layers are investigated, see Fig. 11.

In those layers below 1500 m, the two lines in both subfigures show the same tendency of variation with height −the error value increases with the increase of height. This is because the higher the layer, the more the spread of the GNSS signals, the worse the accuracy of the result. However, in the layers above 1500 m, the two lines show opposite tendencies of variation with height because the higher, the smaller the water vapor density. The smaller water vapor density values in these high layers lead to the small RMSE and large relative errors.





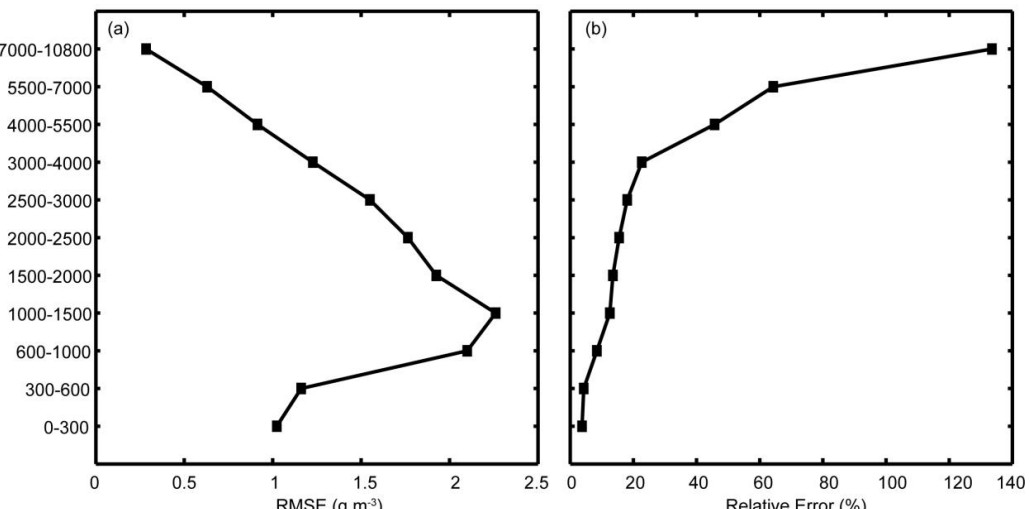

**Figure 11.** (a) Monthly RMSE of (absolute) tomographic errors and (b) mean of relative
tomographic errors in different layers.

**4 Conclusion and outlook**

In this study a new node parameterization approach for determination of a tomographic
field based on the distribution of GNSS signals at the tomographic epoch and also for
discretization of the tomographic field is proposed. The number and the position of the nodes on
each tomographic plane are determined based on the perimeter of the tomographic boundary on
the plane and meshing techniques respectively. Since the tomographic model is tailor-made for
the tomographic field at the epoch, the new approach is applicable to not only GNSS networks
with several stations, but also GNSS networks with few stations, e.g., three stations, which
cannot be solved by conventional approaches. The new approach was tested using GNSS data
from three stations in the Hong Kong Satellite Positioning Reference Station Network during the
period of May, 2015 and its model results were validated by comparing them against radiosonde
data at UTC 0 and UTC 12 from HKKP. Results suggest that the new approach is feasible for a
three-station GNSS network. In addition, monthly statistics of the tomographic results on each
tomographic layer indicated that the size of the tomographic boundary and the magnitude of
water vapor are two critical factors affecting the accuracy of the tomographic result of the layer.

Our future work will be focusing on using unevenly distributed nodes that fit the density
of the GNSS signals.

**Acknowledgements**

This study is supported by the National Natural Science Foundation of China (No.
41730109), the National Natural Science Foundation of China (No. 41774026). The authors
acknowledge the Survey and Mapping Office (SMO) of Lands Department, Hong Kong for
provision of GNSS data from the Hong Kong Satellite Positioning Reference Station Network
(SatRef). We also thank King's Park Observatory for provision of radiosonde data, and the
Department of Earth Atmospheric and Planetary Sciences, MIT for the GAMIT/GLOBK





software. The editor and reviewer team is also highly appreciated for their valuable comments, which makes great improvements in the quality of the paper.

**Data availability**

GNSS data in the RINEX format used for this study can be downloaded from website (http://www.geodetic.gov.hk/smo/gsi/programs/en/GSS/satref/satref.htm). Radiosonde data of King's Park Meteorological Station can be downloaded from website (http://weather.uwyo.edu/upperair/sounding.html).

**Competing interests**

The authors do not have any possible conflicts of interest.

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
