# Peer review of "A new approach for GNSS tomography from a few GNSS stations"

_Atmospheric Measurement Techniques, 2017_

## Referee Comment (RC1) · Anonymous Referee #1 · 23 Apr 2018

General comments:

The proposed in the paper new solution in area of GNSS troposphere tomography is very interesting especially in the case of a small number of GNSS stations in the area covered by the model. It is a fact that with a small number of GNSS stations, a rectangular tomographic model does not reflect the actual GNSS signal paths, and thus the accuracy of the SWD determination is poor. The proposed new approach for determination of tomographic fields in fact roughly an upside-down cone fitting the real distribution of GNSS signals. The proposed solution is clearly written. Also the case studies are documented very well and obtained results confirm assumptions.

Minor comment, question

[Figure]

What is the minimum cut of angle for satellites in proposed new approach? The low satellites are also important for better representation of the distribution of water vapour.

---

## Author Comment (AC1) · 25 Apr 2018

Thank you for your comments on our manuscript. The minimum cut of angle for satellites is 20 degree in new approach. Although the GNSS tomographic observations with low satellite elevations include more water vapour information, the observations, always together with sparse signals, often lead to bad results. This is because the sparse signals increase the tomographic region without comparable observations to calculating the additional parameters in this extended region (Sparse signals were shown in Figure 9(c) and 9 (d) with blue lines in manuscript). In the future, low satellites information may be utilized by non-uniform tomographic model through flexible node parameter approach. For example, the region of sparse signals has a smaller number of nodes than that of other parts of tomographic region.

---

## Referee Comment (RC2) · Anonymous Referee #2 · 25 May 2018

The manuscript is overall well written and truly presents a novel approach for wet refractivity vertical profile reconstruction using GNSS slant wet delays. I don't have any significant major comments, but I would like to ask the authors to adress some mostly minor comments and correct some small language issues.

Comments:

- line 104: I would slightly correct the statement, that ionospheric delay can be cancelled out using an ionosphere-free linear combination. This LC takes care only of the first order effect, not completely everything.

- line 110: term R in given formula (1) is usually explained as a post-fit residual, not as "the unmodeled delay", because it is definetely not given only by the tropospheric delay.

[Figure]

Have you applied post-fit residuals to your SWD used for tomographic reconstruction since you give them in the formula? Have you cleaned them from systematic effects as multipath? See i.e. paper https://www.atmos-meas-tech.net/10/2183/2017/amt-10-2183-2017.pdf, this is a rather important step before using post-fit residuals in slant delays.

- line 146, caption of Fig. 2: please try to correct the form in which dates are given (in the used way the sentence at UTC 0 on 1(DOY 121), ..., in May 2015 is little bit tricky to understand)

- section 3.1: you don't provide any information about the initialization of your tomographic system for the reconstruction, what is an important thing. Have you used some external data as NWM fields, or you used only standard atmosphere water vapor distribution?

- line 316: I would try to correct the sentence where you say that "the number of SWD observations from a tomographic epoch was seven". I would rather write that SWD observations from seven epochs stacked to one tomographic modelling interval were used

- line 318: you present a meteorological situation during May 2015 with total amout of precipitation, but it would be also good to know how the rain events were distributed in time. Were there any severe rains? Or was the precipitation somehow evenly distributed over whole month?

- line 323: I agree that water vapor decreases exponentially with height during standard atmospheric conditions. But there can easily occur inversions of water vapor (wet refractivity) in vertical profile, at some latitudes and time periods they are quite common. Have you analyzed the radiosonde (RS) data on this or not? I mean how you checked if vertical profiles from RS evinced some inversions and if yes, then it would be good to say how often, at which heights, etc.

[Figure]

- section 3.2: do I understand it correctly that you interpolated GNSS tomography results on the RS profile to compare these two techniques? With the term "all RS sampling points" you mean all points were RS provided its measurements? If I understand it correctly then I am not sure if this way of comparison is an optimal one, because RS provides its measurements at slighlty different heights everytime. So I think it would be better to horizontally interpolate GNSS tomography results to the position of RS, and then vertically interpolate RS values to individual layers of your tomographic network. Then you would be comparing all the time the same. Can you comment on this?

- line 352: are you sure that the reason of difference between "the worst and the best results" lies only in the distribution of slant signals used for tomographic reconstruction? Can't there be also an relation with weather conditions or quality of input SWD signals or something else? I would not be so strong in your statement regarding this.

- line 368: you state that statistic values listed in Table 1 are "all small", what means that your "new proposed tomography approach" is feasible". Could you put these values in relation with some other published tomography studies which ideally used a similar territory and a similar season? It could put your numbers into some perspective.

- line 386: I am a little bit surprised with your statements that "... near the ground surface, ... the density of the GNSS signals is very high" - generally the GNSS tomography technique is considered to have troubles to reconstruct water vapor fields well in the boundary layer of troposphere, since the number of slant delays at important low elevation angles is rather limited. What elevation cut-off angle did you use for your GAMIT processing and later in your tomographic reconstruction? Can you comment on this and ideally provide some statistics of how many voxels were penetrated by (how many) signals at various heights? This would support your given statements.

- line 407, table 2: you present a number of outliers, but not the total number of compared values. Were the outlier values included in or excluded from the presented overal monthly statistics?

- I would propose to decrease size of most of the figures in the manuscript

Language correction:

- line 43: use tomographic instead of tomogpaphic

- line 164: I propose to use "... are named pierce points..." instead of current "... are name pierce points..."

---

## Author Comment (AC2) · 4 Jun 2018

Thank you for your comments on our manuscript. These comments are very valuable and helpful for great improvements of the manuscript. In this revised version, each of the comments has been carefully addressed and changes made for corrections are marked in red in the manuscript. The following is the response to the reviewer's comments.

Comment 1:

- line 104: I would slightly correct the statement, that ionospheric delay can be cancelled out using an ionosphere-free linear combination. This LC takes care only of the first order effect, not completely everything.

[Figure]

Response:

The original part "The ionospheric delay can be cancelled out using an ionosphere-free linear combination of dual-frequency observations." has been replaced with "The first order ionospheric delay was eliminated using the so-called ionosphere-free linear combination of dual-frequency observations" in lines 104-106.

Comment 2:

- line 110: term R in given formula (1) is usually explained as a post-fit residual, not as "the unmodeled delay", because it is definetely not given only by the tropospheric delay. Have you applied post-fit residuals to your SWD used for tomographic reconstruction since you give them in the formula? Have you cleaned them from systematic effects as multipath? See i.e. paper https://www.atmos-meas-tech.net/10/2183/2017/amt-10-2183-2017.pdf, this is a rather important step before using post-fit residuals in slant delays.

Response:

(1) We actually applied post-fit residuals to SWD used for tomographic reconstruction in this paper. In one satellite-receiver (e.g. PRN03 and HKSC), we removed the residuals exceeding 2.5 times the standard deviation and also their corresponding observations from our tomographic procedure and then computed means were subtracted from the other post-fit residuals to clean observation from systematic effects.

(2) We have read the paper "Inter-technique validation of tropospheric slant total delays". The method about obtaining the cleaned residuals is a great way to extract the anisotropic components especially the stacking technique. In our study, the area of our interest ranges from 113.749° E to 114.474° E in the longitudinal direction, from 22.115° N to 22.651° N in the latitudinal direction. It is a small area for the GNSS tomography (All SWD observations in the $1\times1°$bin). Therefore, we used another approach in similar way − cleaning of post-fit residuals based on the residuals from one

satellite-receiver, to some extent, ensures the residuals under the same conditions (i.e. the residuals from the same satellite and receiver). The stacking technique based on generating elevation and azimuth correction maps is a good way to clean the residuals. We will research it in the future.

The relevant changes made are as follows:

The original part "R is the unmodeled delay;" has been replaced with "R is the post-fit residuals and in one satellite-receiver, the residuals exceeding 2.5 times the standard deviation were removed and then computed means were subtracted from the remaining residuals to clean observation from systematic effects;" in lines 111-113.

Comment 3:

- line 146, caption of Fig. 2: please try to correct the form in which dates are given (in the used way the sentence at UTC 0 on 1(DOY 121), ..., in May 2015 is little bit tricky to understand)

Response:

The original part "at UTC 0 on 1(DOY 121), 16 (DOY 136) and 31 (DOY 151) in May 2015." has been replaced with "with the sampling rate of 30 seconds at UTC 0 on 1, 16 and 31 May, 2015." in line 146.

Comment 4:

- section 3.1: you don't provide any information about the initialization of your tomographic system for the reconstruction, what is an important thing. Have you used some external data as NWM fields, or you used only standard atmosphere water vapor distribution?

Response:

We did not use any information for the initialization of tomographic system. In the inverse process, the zero vector as the initial value of the unknown parameters was

used based on the algebraic reconstruction technique (ART).

Comment 5:

- line 316: I would try to correct the sentence where you say that "the number of SWD observations from a tomographic epoch was seven". I would rather write that SWD observations from seven epochs stacked to one tomographic modelling interval were used

Response:

The original part "the number of SWD observations from a tomographic epoch was seven" has been replaced with "SWD observations from seven epochs stacked to one tomographic modelling interval were used" in lines 315-316.

Comment 6:

- line 318: you present a meteorological situation during May 2015 with total amount of precipitation, but it would be also good to know how the rain events were distributed in time. Were there any severe rains? Or was the precipitation somehow evenly distributed over whole month?

Response:

The original part "The reason for the selection of data from May 2015 is that its monthly total rainfall was 513.0 mm, a 68% larger than the normal level of 304.7 mm." has been replaced with "The reason for the selection of data from May 2015 is that its monthly total rainfall was 513.0 mm, a 68% larger than the normal level of 304.7 mm. The weather in Hong Kong was hot on the first few days of the month. After a cloudy but relatively rain-free day on 8 May, another trough of low pressure brought heavier showers and thunderstorms to Hong Kong on 9-10 May. Two rainstorm episodes on 20 and 23 May brought rain to most parts of the Hong Kong. Another rapidly developed rainstorm was on 26 May. The weather improved gradually with sunny periods on 28-30 May. However, the weather turned cloudy again with isolated showers and thunderstorms on 31

May." in lines 317-324.

Comment 7:

- line 323: I agree that water vapor decreases exponentially with height during standard atmospheric conditions. But there can easily occur inversions of water vapor (wet refractivity) in vertical profile, at some latitudes and time periods they are quite common. Have you analyzed the radiosonde (RS) data on this or not? I mean how you checked if vertical profiles from RS evinced some inversions and if yes, then it would be good to say how often, at which heights, etc.

Response:

We have not analyzed the radiosonde (RS) data on inversions of water vapor. But we also concerned about this phenomenon. In fact, the inversions of water vapor occurred in the heavy rain event on 26 May has been detected by our GNSS tomography results. Some possible assumptions were proposed for this phenomenon due to lack of the radiosonde (RS) data for reference. As shown in Figure 1, we divided the whole process of rain into three periods (i.e., Figure 1(a): 8 am to 9 am, Figure 1(b): 9 am to 10 am and Figure 1(c):10 am to 11 am). In Figure 1(a), HKKP has received just 1 millimeters of rain but the water vapor below 3000m was increased (the inversion of water vapor) during 8:20 am to 8:40 am and then the inversion lifted to between 3000m and 5000m during 8:40 am to 9:00 am. In Figure 1(b), there was 20 millimeters of rain with the inversion declined from 3000m to 1000m during 9 am to 10 am. In Figure 1(c), with the 40 millimeters of rain, the inversion gradually fades away. We deduced that the vertical motion of water vapor, especially ascending motion, is an initial condition of the heavy rain event. It is note that the above results were not validated due to the lack of references and sufficient tests. But we all interested in this phenomenon, which is also one of the important objectives in the future study.

Comment 8:

- section 3.2: do I understand it correctly that you interpolated GNSS tomography results on the RS profile to compare these two techniques? With the term "all RS sampling points" you mean all points were RS provided its measurements? If I understand it correctly then I am not sure if this way of comparison is an optimal one, because RS provides its measurements at slighlty different heights everytime. So I think it would be better to horizontally interpolate GNSS tomography results to the position of RS, and then vertically interpolate RS values to individual layers of your tomographic network. Then you would be comparing all the time the same. Can you comment on this?

Response:

Yes, we interpolated GNSS tomography results on the RS profile to compare these two techniques. The RS data in our study is used for the reference of the tomographic results, so we avoid adjusting the original data of RS. The other reason is that the RS data also exists the potential error especially in rainy day. Comparison only between the layers of tomographic results and corresponding vertically interpolate RS values may increases the influence of RS error due to insufficient results of comparison. Finally, GNSS tomography is a technique for building the water vapor distribution in three dimensions. Therefore, its performance at different heights is also important for validation.

Comment 9:

- line 352: are you sure that the reason of difference between "the worst and the best results" lies only in the distribution of slant signals used for tomographic reconstruction? Can't there be also an relation with weather conditions or quality of input SWD signals or something else? I would not be so strong in your statement regarding this.

Response:

You're right. The results have an relation with weather conditions or quality of input SWD signals or something else. The distribution of slant signals used for tomographic

reconstruction is one of possible reasons for the accuracy of the model results. The sparse signals would reduce the accuracy in three-stations model due to there are insufficient observations and relatively more unknown parameters (caused by the sparse signals) for GNSS tomography. However, this situation did not exist in GNSS network since the sufficient observations and uniform distribution of slant signals were used in GNSS tomographic model.

The relevant changes made are as follows:

The original part "which is the reason for the poor accuracy of the model results." has been replaced with "which is one of possible reasons for the poor accuracy of the model results." in lines 363-364.

Comment 10:

-line 368: you state that statistic values listed in Table 1 are "all small", what means you're your "new proposed tomography approach" is feasible". Could you put these values in relation with some other published tomography studies which ideally used a similar territory and a similar season? It could put your numbers into some perspective.

Response:

One published tomography studies was introduced to support "new proposed tomography approach" is feasible".

The relevant changes made are as follows:

The original part "The statistics of the model results at both epochs together in the whole month are presented in Table 1. The three values listed in the table are all small, meaning that the new approach for a few GNSS stations, such as three stations, is feasible. " has been replaced with "In Table 1, the statistics of the model results at both epochs together in the whole month are compared with that of the adaptive node parameterization approach (ANP) (Ding et al., 2018) during the same periods. Unlike the results of new approach are based on three stations of SatRef, 17 stations of

SatRef are used to estimate the results of the ANP. The RMSE and IQR values of new approach are similar to that of the ANP, meaning that the new approach for a few GNSS stations, such as three stations, is feasible. But in terms of the Bias, the new approach has a poor performance." in lines 370-376. Table 1. Monthly statistics of new approach and ANP Statistic RMSE (g m-3) Bias (g m-3) IQR (g m-3) New approach 1.477 0.239 1.430 ANP 1.216 -0.012 1.678 Reference: Ding N, Zhang S B, Wu S Q, et al. Adaptive Node Parameterization for Dynamic Determination of Boundaries and Nodes of GNSS Tomographic Models[J]. Journal of Geophysical Research Atmospheres, 2018, 123(4).

Comment 11:

- line 386: I am a little bit surprised with your statements that "... near the ground surface, ... the density of the GNSS signals is very high" - generally the GNSS tomography technique is considered to have troubles to reconstruct water vapor fields well in the boundary layer of troposphere, since the number of slant delays at important low elevation angles is rather limited. What elevation cut-off angle did you use for your GAMIT processing and later in your tomographic reconstruction? Can you comment on this and ideally provide some statistics of how many voxels were penetrated by (how many) signals at various heights? This would support your given statements.

Response:

1) You're right. The GNSS tomography technique is considered to have troubles to reconstruct water vapor fields well in the boundary layer of troposphere. It is not only due to the number of slant delays at important low elevation angles is rather limited, but also due to the value of water vapor above 4000m is reduced rapidly with the increase of height (Jiang et al., 2014). The latter one is the main reason for the poor performance in the boundary layer of troposphere since the value of water vapor in this layer is too small to be estimated. Elevation cut-off angle is 10 degree for the GAMIT processing and the minimum cut of angle is 20 degree in tomographic reconstruction. Although the GNSS tomographic observations with low satellite elevations include more water

vapour information, the observations, always together with sparse signals, often lead to bad results. This is because the sparse signals increase the tomographic region without comparable observations to calculating the additional parameters in this extended region (Sparse signals were shown in Figure 9(c) and 9 (d) with blue lines in manuscript).

2) There are about 20 nodes were penetrated by about 106 signals on the bottom of the tomographic region (300m) and 28 nodes were penetrated by the same number of signals on top of tomographic region (10800m). However, the area of tomographic boundaries on the bottom is about 64 km2 (i.e., about 1.7 signals per square kilometer) and that on the top is 1550 km2 (i.e., about 0.07 signals per square kilometer). Therefore, the density of the GNSS signals is very high near the ground surface. The above content was added in the paper for supporting our given statements.

The relevant changes made are as follows:

The original part "and the density of the GNSS signals is very high, resulting in relative high accuracy;" has been replaced with "and there are about 20 nodes were penetrated by about 106 signals on the bottom of the tomographic region (300m) and 28 nodes were penetrated by the same number of signals on the top of tomographic region (10800m). However, the area of tomographic boundaries on the bottom is about 64 km2 (i.e., about 1.7 signals per square kilometer) and that on the top is 1550 km2 (i.e., about 0.07 signals per square kilometer). Therefore, the density of the GNSS signals is very high near the ground surface, which results in relative high accuracy;" in lines 393-399.

Reference:

Jiang P, Ye S R, Liu Y Y, et al. Near real-time water vapor tomography using ground-based GPS and meteorological data: long-term experiment in Hong Kong[J]. Annales Geophysicae, 2014, 32(8):911-923.

Comment 12:

- line 407, table 2: you present a number of outliers, but not the total number of compared values. Were the outlier values included in or excluded from the presented overall monthly statistics?

Response:

The total number of compared values is 2790. The outlier values were included in the presented overall monthly statistics. The relevant changes made are as follows: The original part "Table 2 lists all the above characteristic values." has been replaced with "Table 2 lists all the above characteristic values of all errors (the total number of errors is 2790)." in lines 413-414.

Comment 13:

I would propose to decrease size of most of the figures in the manuscript.

Response: Amended as suggested

Comment 14:

Language correction: - line 43: use tomographic instead of tomogpaphic - line 164: I propose to use "... are named pierce points..." instead of current "... are name pierce points..."

Response: Amended as suggested

Please also note the supplement to this comment:
https://www.atmos-meas-tech-discuss.net/amt-2017-426/amt-2017-426-AC2-supplement.zip

[Figure]

Figure 1. Water vapor profile of heavy rain event on 26 May (Response of Comment 7)

**Fig. 1.** Figure 1. Water vapor profile of heavy rain event on 26 May (Response of Comment 7)